# Jointly Modelling Uncertainty and Diversity for Active Molecular Property Prediction

**Kuangqi Zhou**[1]*  **Kaixin Wang**[1]  **Jian Tang**[2,3]  **Jiashi Feng**[4]

**Bryan Hooi**[1]  **Peilin Zhao**[5]  **Tingyang Xu**[5]†  **Xinchao Wang**[1]†

{kzhou,kaixin.wang}@u.nus.edu   jian.tang@hec.ca
jshfeng@gmail.com   dcsbhk@nus.edu.sg
{masonzhao,tingyangxu}@tencent.com   xinchao@nus.edu.sg

[1]National University of Singapore   [2]HEC Montreal
[3]MILA   [4]ByteDance   [5]Tencent AI Lab

## Abstract

Molecular property prediction is a fundamental task in AI-driven drug discovery. Deep learning has achieved great success in this task, but relies heavily on abundant annotated data. However, annotating molecules is particularly costly because it often requires lab experiments conducted by experts. Active Learning (AL) tackles this issue by querying (*i.e.*, selecting) the most valuable samples to annotate, according to two criteria: *uncertainty* of the model and *diversity* of data. Combining both criteria (a.k.a. hybrid AL) generally leads to better performance than using only one single criterion. However, existing best hybrid methods rely on some trade-off hyperparameters for balancing uncertainty and diversity, and hence need to carefully tune the hyperparameters in each experiment setting, causing great annotation and time inefficiency. In this paper, we propose a novel AL method that jointly models uncertainty and diversity without the trade-off hyperparameters. Specifically, we model the joint distribution of the labeled data and the model prediction. Based on this distribution, we introduce a Minimum Maximum Probability Querying (MMPQ) strategy, in which a single selection score naturally captures how the model is uncertain about its prediction, and how dissimilar the sample is to the currently labeled data. To model the joint distribution, we adapt the energy-based models to the non-Euclidean molecular graph data, by learning chemically-meaningful embedding vectors as the proxy of the graphs. We perform extensive experiments on binary classification datasets. Results show that our method achieves superior AL performance, outperforming existing methods by a large margin. We also conduct ablation studies to verify different design choices.

## 1 Introduction

AI-driven drug discovery is an important application of machine learning. In drug discovery pipeline, a fundamental step is to use computational methods to predict the molecular properties (*e.g.*, toxicity and binding specificity) of candidate compounds [1, 2]. Recently, deep learning models have achieved great success in molecular property prediction [1, 3–5], but their high performance relies on a large

---

*This work is done when Kuangqi Zhou works as an intern in Tencent AI Lab.
†Co-Corresponding Authors

K. Zhou et al., Jointly Modelling Uncertainty and Diversity for Active Molecular Property Prediction. *Proceedings of the First Learning on Graphs Conference (LoG 2022)*, PMLR 198, Virtual Event, December 9–12, 2022.

amount of annotation. However, annotating molecules is particularly time-consuming and costly, since it often requires lab experiments or complex computation [3, 6].

One promising way to alleviate this problem is Active Learning (AL) [7], which aims at finding a strategy for iteratively querying (*i.e.*, selecting) the most valuable data samples to annotate, so as to maximize model performance under a low annotation budget. AL strategies query samples mainly based two criteria: *uncertainty* of the model [8], and *diversity* of queried data [9]. Strategies taking into account both criteria (a.k.a. hybrid strategies) are recently shown to outperform methods based on only uncertainty or diversity in many learning tasks [10–12]. Existing best hybrid methods generally rely on some *trade-off hyperparameters* for balancing uncertainty and diversity [11–15]. For example, WAAL [12] requires manually-tuned coefficients to obtain a weighted sum of its uncertainty and diversity terms. EADA [13] relies on two selection ratios for its two-step selection process. These trade-off hyperparameters are crucial to the AL performance and hence need to be carefully tuned for each experiment setting.

However, tuning trade-off hyperparameters can cause substantial inefficiency in AL. For one thing, since these hyperparameters have a large influence on the outcome of corresponding AL strategies, the selected samples under different choices of the hyperparameters often vary a lot, and thus the total annotation cost needed for tuning will greatly exceed the budget. For another, the tuning process can take a long time, since each AL experiment iterates between query selection and model (re)training for several rounds. One possible way of addressing this drawback is to tune the hyperparameters in advance on a dataset with small annotation cost. However, this would be problematic when the dataset used for tuning is not representative. Indeed, for molecular property prediction, finding a representative dataset for tuning would be difficult, since the properties of interest vary a lot.

In this paper, we propose a novel AL strategy that naturally takes into account uncertainty and diversity without the need of trade-off hyperparameters. Our strategy is based on a joint distribution $q(x, y) \triangleq p(y|x)p(x)$, which contains information about both uncertainty and diversity: $p(y|x)$ is the prediction distribution of the model (with input $x$ and prediction $y$), which is widely used to define uncertainty metrics [7, 8, 16]; $p(x)$ is the density of the currently annotated data, which is shown to be useful for identifying samples that can effectively increase data diversity [11, 12, 17].

Specifically, our strategy operates by first maximizing $q(x, y)$ via varying $y$, and then minimizing $\max_y q(x, y)$ via varying $x$. We thus name our strategy Minimum Maximum Probability Querying (MMPQ). Importantly, we show that the selection score of MMPQ can be viewed as the product of two terms — the first term leads to samples on which the model has low prediction confidence, while the second favors samples that are dissimilar to labeled data. In this way, the selected samples are naturally those that the model is most uncertain about, while at the same time being able to increase the data diversity.

For modelling the joint distribution, we propose to use an Energy-Based Model (EBM) [18, 19], since it can explicitly output the desired probability value. For training the EBM, we need to tackle one key challenge in our setting: the variable $x$ in the joint distribution has a non-Euclidean data structure (*i.e.*, a molecule graph), which renders the commonly-used EBM training scheme inapplicable [19, 20]. To address this challenge, we take a learned embedding vector $z$ as a proxy of the non-Euclidean input $x$, which allows us to train the EBM on $z$ and $y$ with the commonly-used EBM training scheme. Specifically, inspired from [21, 22], we learn the embeddings by training an autoencoder to reconstruct the input SMILES strings, which is an expert-defined sequence representation of molecules. The EBM is trained by Denoising Score Matching (DSM) [20, 23], *i.e.*, to learn the "Stein score" [24] of $q(x, y)$, which has been shown to be an efficient and robust EBM training scheme.

To evaluate our MMPQ strategy, we apply it to actively train a commonly-used Graph Neural Network (GNN) [4] on various benchmark datasets of binary molecular property classification. Extensive results show that MMPQ enables the GNN to achieve high performance with a limited annotation budget, significantly outperforming other competitive AL methods. We conduct ablation studies to verify different design choices of our method. In particular, we show that the uncertainty and diversity terms make complementary contributions to good performance: the diversity term is important in early iterations of AL, while the uncertainty term is essential in later iterations.

## 2 Related Works

**Molecular property prediction** is a critical step in drug discovery [1, 2, 5]. Traditional methods (*e.g.*, based on density function theory [25]) are too slow to be applied in practice. To resolve this problem, deep learning methods [1, 3, 4, 26, 27] have been widely proposed, which can be categorized into two types: (1) descriptor-based methods [26, 27] that represent the input molecules as expert-crafted molecular descriptors (*e.g.*, fingerprints [28]), and (2) GNN-based methods [1, 3, 4] that directly take molecule graphs as input. As found in [1, 5], GNNs generally outperform descriptor-based methods, and thus this work focuses on GNN-based molecular property prediction.

**Active learning** improves annotation efficiency by iteratively querying samples based on two criteria: uncertainty of the task learner [8, 16, 29, 30], and/or diversity of queried data [9, 17, 21]. Uncertainty-based methods define various uncertainty metrics for querying data [8, 10, 16, 30], while diversity-based methods aim to find a representative subset of the whole dataset by querying diverse samples [9, 21]. Compared to using only uncertainty or diversity, recent works find that combining the two criteria (a.k.a. hybrid methods) leads to better performance [10, 11, 13–15]. However, existing hybrid approaches generally need to balance uncertainty and diversity via some trade-off hyperparameters. For example, WAAL [12] uses manually-tuned coefficients to weight its uncertainty term and diversity term. EADA [13] adopts a two-stage querying approach, where each stage requires a prefixed selection ratio. We note that EADA also trains an EBM for active selection. Our method differs from EADA mainly in 3 aspects. First, the motivation of EADA to adopt EBMs is to identify out-of-distribution samples, while we use EBMs for modeling the distribution capturing both uncertainty and diversity. Second, they train their EBM via contrastive divergence [31], while we use denoising score matching [20]. Third, they need two separate selection steps with different selection scores, and require two hyperparameters to trade off between uncertainty and diversity, while we only have one single selection score that naturally captures both uncertainty and diversity.

Apart from the above, some other hybrid methods rely on trade-off hyperparameters during their model training process [11, 14, 15]. We note that there is an existing hybrid strategy, BADGE [10], that is also free from trade-off hyperparameters like ours. However, BADGE [10] assumes that task learner's prediction is a faithful proxy of the ground-truth label. This may not hold in early AL iterations on (typically) small molecule datasets, since the task learner would be inaccurate due to limited training data.

**Energy-based models** are a class of powerful methods of explicit generative modeling. Recently, some works [32–35] leverage EBMs for modeling molecular data. To tackle difficulties caused by the discrete nature of molecule graphs, [32] leverages a dequantization technique, and [33] designs a diffusion process based on stochastic differential equations. Different from [32, 33], we propose to train our EBM on a continuous embedding space of molecules. On the other hand, [34, 35] focuses on molecule conformation generation, which is essentially a continuous problem, since the conformation of a molecule is represented by the 3D space coordinates of its atoms.

## 3 Preliminaries

**Problem Setting.** We consider batch-mode pool-based active learning [7], a practical setting for deep models. In each AL round, a batch of samples from the unlabeled pool $\mathcal{D}_U$ are queried according to a strategy, annotated by an oracle, and added to the labeled pool $\mathcal{D}_L$. The updated $\mathcal{D}_L$ is then used to train the task learner. A more formal description of this setting is in Appx. A.1.

**Notations.** A molecule is represented as a graph $G = (V, E)$, with nodes $V$ and edges $E$ corresponding to atoms and chemical bonds. As in [4, 36–38], we are interested in $n$ binary molecular properties (*e.g.*, toxicity), which are denoted by a label vector $\mathbf{y} = (y_1, \cdots, y_n) \in \{0, 1\}^n$, where $y_i = 1$ or $0$ means the molecule has the $i$-th property or not. A task learner $h(\cdot)$ is trained to predict the properties. The $i$-th output of the task learner, $h(G)_i$, specifies a distribution $p(y_i|G)$ over the predicted label of the $i$-th property of $G$, which is essentially a Bernoulli distribution with success probability $h(G)_i$ (denoted as $\text{Ber}(h(G)_i)$).

**Energy-Based Models.** EBMs [18] specify probability density or mass functions as follows:

$$p_\theta(\mathbf{x}) = \frac{\exp(-E_\theta(\mathbf{x}))}{Z_\theta},\tag{1}$$

where $\mathbf{x} \in \mathbb{R}^D$ is a random sample, $E_\theta(\mathbf{x})$ is the *energy function* with learnable parameters $\theta$, and $Z_\theta = \int \exp(-E_\theta(\mathbf{x})) \, \mathrm{d}\mathbf{x}$ is a normalizing constant. By learning $\theta$, we can use an EBM to approximate a real data distribution, *i.e.*, $p_\theta \approx p_{\text{data}}$.

**Denoising Score Matching.** DSM [23, 39] is an efficient approach for training EBMs. Here, the "(Stein) score" of a distribution $f(\mathbf{x})$ is defined as the log-probability's first-order gradient function w.r.t. $\mathbf{x}$, *i.e.*, $\nabla_\mathbf{x} \log f(\mathbf{x})$. DSM first disturbs $p_{\text{data}}(\mathbf{x})$ with a pre-defined noise distribution $p_N(\tilde{\mathbf{x}}|\mathbf{x})$, and then trains the EBM via

$$\mathbb{E}_{\substack{\mathbf{x} \sim p_{\text{data}}(\mathbf{x}) \\ \tilde{\mathbf{x}} \sim p_N(\tilde{\mathbf{x}}|\mathbf{x})}} \left[ \frac{1}{2} \left\| \nabla_\mathbf{x} \log p_\theta(\tilde{\mathbf{x}}) - \nabla_\mathbf{x} \log p_N(\tilde{\mathbf{x}}|\mathbf{x}) \right\|_2^2 \right]. \tag{2}$$

With a proper $p_N(\tilde{\mathbf{x}}|\mathbf{x})$, we can easily obtain $\nabla_\mathbf{x} p_N(\tilde{\mathbf{x}}|\mathbf{x})$ [39].

## 4 Method

### 4.1 The Minimum-Maximum-Probability Query Strategy

Our proposed query strategy is based on the *joint distribution* of two key probability distributions used in existing works. The first is the prediction distribution of the task learner, *i.e.*, $p(y = \hat{y}|G)$ (abbreviated as $p(\hat{y}|G)$), which is widely used to define different uncertainty metrics [7, 8, 16]. The second is the distribution of currently labeled pool $\mathcal{D}_L$, denoted as $p_L(G)$. As shown in [11, 12, 17, 21], $p_L(G)$ is useful for identifying samples that are dissimilar to the labeled ones, and annotating these samples effectively increases data diversity. Inspired by these works, we propose to model the joint distribution of $p(y = \hat{y}|G)$ and $p_L(G)$.

Formally, let $q(G, \hat{\mathbf{y}})$ denote the joint distribution:

$$q(G, \hat{\mathbf{y}}) \triangleq p(\hat{\mathbf{y}}|G)p_L(G), \tag{3}$$

where we use the boldface $\hat{\mathbf{y}}$ because we may be interested in more than 1 tasks. Note that $\hat{\mathbf{y}}$ is a random variable following $p(\hat{\mathbf{y}}|G)$, not the ground-truth label of $G$.

Then, we perform active selection by first maximizing $q(G, \hat{\mathbf{y}})$ via varying $\hat{\mathbf{y}}$ for each single $G$, and then selecting a batch of $G$ that minimizes the obtained $\max_{\hat{\mathbf{y}}} q(G, \hat{\mathbf{y}})$. Denote the selected batch as $\mathcal{B} = \{G_1, \cdots, G_b\}$. Our strategy is formalized as:

$$\mathcal{B} = \underset{G_1, \cdots, G_b \in \mathcal{D}_U}{\arg\min} \left( \max_{\hat{\mathbf{y}}} q(G_1, \hat{\mathbf{y}}), \cdots, \max_{\hat{\mathbf{y}}} q(G_b, \hat{\mathbf{y}}) \right). \tag{4}$$

We name our strategy Minimum-Maximum-Probability Querying (MMPQ). The whole active learning process with MMPQ strategy is summarized in Appx. A.4.

#### 4.1.1 MMPQ as a Tuning-Free Hybrid Strategy

Here we show that MMPQ naturally captures both uncertainty of the task learner and diversity w.r.t. the whole data space in a tuning-free manner. First, from Eqn. (4) and Eqn. (3), we can see that the selection score of MMPQ is:

$$\max_{\hat{\mathbf{y}}} q(G, \hat{\mathbf{y}}) = \left( \max_{\hat{\mathbf{y}}} p(\hat{\mathbf{y}}|G) \right) p_L(G). \tag{5}$$

Then, let $p^M = \max_{\hat{\mathbf{y}}} p(\hat{\mathbf{y}}|G)$, and it can be seen that the MMPQ strategy essentially selects data with smaller $p^M$ and smaller $p_L(G)$.

▶ **Uncertainty.** Smaller $p^M$ corresponds to samples that the task learner is less confident about. Specifically, let $\hat{\mathbf{y}}^* = (\hat{y}_1^*, \cdots, \hat{y}_n^*)$ denote the prediction that achieves $p^M$ (*i.e.*, $p^M = p(\hat{\mathbf{y}}^*|G)$), and $\hat{\mathbf{y}}' = (\hat{y}_1', \cdots, \hat{y}_n')$ denote any other prediction that is different from $\hat{\mathbf{y}}^*$. Note that, since $p(\hat{\mathbf{y}}^*|G) + \sum_{\hat{\mathbf{y}}' \in \{0,1\}^n, \hat{\mathbf{y}}' \neq \hat{\mathbf{y}}^*} p(\hat{\mathbf{y}}'|G) = 1$, smaller $p^M$ means that $p^M$ is closer to the second-largest (and all other) predictions, implying that the task learner is more uncertain about its prediction on molecule $G$.

▶ **Diversity.** Smaller $p_L(G)$ means that $G$ lies in low-density regions of the distribution of currently labeled data, and hence is dissimilar to the labeled data. Thus, querying those with small $p_L(G)$ increases the diversity of the obtained labeled pool $\mathcal{D}_L^t$ [11, 12, 17, 21].

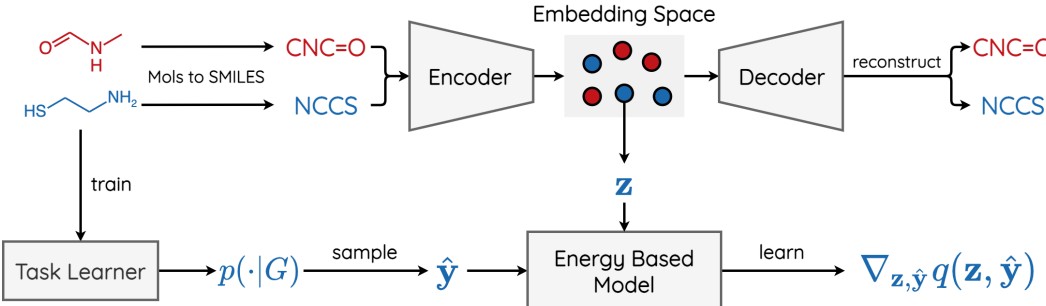

**Figure 1:** Model design and data flow. Colored in blue are inputs and outputs of the labeled pool and corresponding objective. Those corresponding to the unlabeled pool are in red.

Based on above reasoning, samples with lowest selection score (*i.e.*, those taken by the $\arg\min$ operation in Eqn. (4)) are naturally those the model is most uncertain about, while at the same time being able to increase data diversity. As such, MMPQ does not need a hyperparameter to trade off between uncertainty and diversity.

### 4.1.2   Implementation of MMPQ

Since MMPQ is based on the value of $q(G, \hat{\mathbf{y}})$, we need to model $q(G, \hat{\mathbf{y}})$ with an *explicit* deep generative model. In particular, we instantiate an Energy-Based Model (EBM) using a neural network, since EBMs have been shown to be quite expressive and stable in distribution modelling [39, 40]. Formally, $q(G, \hat{\mathbf{y}})$ is modelled by

$$q(G, \hat{\mathbf{y}}) = \frac{\exp(-E(G, \hat{\mathbf{y}}))}{Z}, \tag{6}$$

where $E(G, \hat{\mathbf{y}})$ is the energy value given by the EBM, and $Z$ is a normalizing constant.

In this subsection, we focus on how to implement MMPQ with the EBM, and thus here we assume that the EBM is already trained and fixed. Model design and training of the EBM will be presented later in Sec. 4.2 and Sec. 4.3.

From Eqn. (6), we have

$$
\begin{aligned}
&\arg\min_{G} \left( \max_{\hat{\mathbf{y}}} q(G, \hat{\mathbf{y}}) \right) \\
=&\arg\min_{G} \left( \max_{\hat{\mathbf{y}}} \left( \log q(G, \hat{\mathbf{y}}) + \log Z \right) \right) \\
=&\arg\min_{G} \left( -\min_{\hat{\mathbf{y}}} E(G, \hat{\mathbf{y}}) \right) \\
=&\arg\max_{G} \left( \min_{\hat{\mathbf{y}}} E(G, \hat{\mathbf{y}}) \right).
\end{aligned}
\tag{7}
$$

This reveals that we can implement MMPQ based on the learned energy values, without the need to calculate the normalizing constant $Z$.

One may argue that, $\min_{\hat{\mathbf{y}}} E(G, \hat{\mathbf{y}})$ would be difficult to compute for large $n$, as it involves all $2^n$ possible combinations of $(\hat{y}_1, \cdots, \hat{y}_n)$. We show in Appx. A.2 that, based on the conditional independence assumption of labels [41], $\min_{\hat{\mathbf{y}}} E(G, \hat{\mathbf{y}})$ can be computed in a task-wise manner.

### 4.2   Model Design

Designing an EBM for learning $q(G, \hat{\mathbf{y}})$ is not trivial, since the two variables have different data structures: $G$ is an attributed graph, while $\hat{\mathbf{y}}$ is a vector. Moreover, learning EBMs for attributed graphs is itself a challenging open problem, due to the non-Euclidean and discrete nature [32, 42].

To address the above issues, we propose to embed molecules graphs $G$ into a learned embedding space, and then build the EBM model on $\hat{\mathbf{y}}$ and embeddings $\mathbf{z}$ (see Fig. 1). Inspired by Sinha *et al.* [21], we

learn the space by training an Auto-Encoder (AE) to reconstruct its inputs. However, due to graph isomorphism, directly reconstructing molecule graphs is difficult [22, 43, 44]. We thus propose to train the AE to reconstruct the molecules' SMILES strings [45], as shown in Fig. 1. SMILES is a sequence representation of molecules, where the sub-strings correspond to chemically-meaningful substructures in molecules (*e.g.*, functional groups). Such a sequence-based reconstruction task enables the auto-encoder to learn molecules embeddings without struggling to reconstruct graphs.

Formally, let $\mathrm{Enc}(\cdot)$ and $\mathrm{Dec}(\cdot)$ denote the encoder and decoder respectively, and let $\mathrm{Sml}(\cdot)$ denote the operation of retrieving the SMILES string of a molecule (which can be easily pre-computed using open-sourced cheminformatics libraries). Then, for a molecule $G$, the ground-truth and the reconstructed SMILES strings are

$$S \triangleq \mathrm{Sml}(G), \quad \hat{S} \triangleq \mathrm{Dec}(\mathrm{Enc}(\mathrm{Sml}(G))). \tag{8}$$

For learning high-quality embeddings, we use both labeled and unlabeled data to train the AE:

$$L_{\mathrm{rec}} = \mathbb{E}_{G \in \mathcal{D}_L}\big[d(S, \hat{S})\big] + \mathbb{E}_{G' \in \mathcal{D}_U}\big[d(S', \hat{S}')\big], \tag{9}$$

where $d(\cdot, \cdot)$ is a distance between sequence pairs.

In the rest of this paper, we take $\mathbf{z} \triangleq \mathrm{Enc}(\mathrm{Sml}(G))$ as a proxy of $G$ in some cases, and use $\mathbf{x}$ to denote the tuple $(\mathbf{z}, \hat{\mathbf{y}})$, which is implemented by concatenating $\mathbf{z}$ and $\hat{\mathbf{y}}$.

Following previous works [39, 40], we instantiate the EBM as a "score net" $s_\theta(\mathbf{x})$, which learns the score of the target distribution $q(\mathbf{x})$, *i.e.*, $\nabla_{\mathbf{x}} \log q(\mathbf{x})$. When $s_\theta(\mathbf{x})$ is trained, we use summation to approximate integral over $\nabla_{\mathbf{x}} \log q(\mathbf{x})$ (see Appx. A.3). An alternative choice is to approximate the energy function $E_\theta(\mathbf{x})$, which however is more difficult than modeling the score, as experimentally shown in Sec. 5.3.3.

### 4.3 Model Training

We train the EBM $s_\theta(\mathbf{x})$ via denoising score matching. With the noise $p_N(\tilde{\mathbf{x}}|\mathbf{x}) = \mathcal{N}(\tilde{\mathbf{x}}|\mathbf{x}, \sigma^2, I)$, we have $\nabla_{\mathbf{x}} p_N(\tilde{\mathbf{x}}|\mathbf{x}) = -\frac{\tilde{\mathbf{x}} - \mathbf{x}}{\sigma^2}$ [39]. Then, the DSM objective is:

$$L_{\mathrm{DSM}} = \mathbb{E}_{\substack{\mathbf{x} \in \mathcal{D}_L, \\ \tilde{\mathbf{x}} \sim \mathcal{N}(\tilde{\mathbf{x}}|\mathbf{x}, \sigma)}} \left[ \left\| s_\theta(\tilde{\mathbf{x}}) + \frac{\tilde{\mathbf{x}} - \mathbf{x}}{\sigma^2} \right\|_2^2 \right], \tag{10}$$

where we slightly abuse the notation, using $\mathbf{x} \in \mathcal{D}_L$ to denote $\mathbf{x} \in \{(\mathrm{Enc}(\mathrm{Sml}(G)), \hat{\mathbf{y}})|G \in \mathcal{D}_L\}$.

Note that the second term in the target distribution (Eqn. (3)) is the density of the labeled data only. Therefore, in Eqn. (10), we calculate $L_{\mathrm{DSM}}$ only on the labeled pool $\mathcal{D}_L$ (*cf*. the reconstruction objective in Eqn. (9)).

One challenge of calculating $L_{\mathrm{DSM}}$ is that it requires $(G, \hat{\mathbf{y}})$ pairs i.i.d. sampled from $q(G, \hat{\mathbf{y}})$, but we do not have such samples at hand. To address this challenge, we propose a two-step sampling method: first, randomly pick $G$ from $\mathcal{D}_L$; then draw a sample $\hat{\mathbf{y}} = \{\hat{y}_1, \cdots, \hat{y}_n\}$ from $p(\hat{\mathbf{y}}|G)$, which can be implemented by drawing a $\hat{y}_i \sim \mathrm{Ber}(h(G)_i)$ for all $i$ (under the conditional-independence assumption of labels).

The EBM and the AE are jointly trained via

$$L_{\mathrm{joint}} = L_{\mathrm{DSM}} + L_{\mathrm{rec}}. \tag{11}$$

Pseudo code of model training process is summarized in Appx. A.4.

## 5 Experiments

### 5.1 Experiment Setup

We run experiments under the batch-mode pool-based AL setting (elaborated in Appx. A.1). The labeled pool is initialized by randomly selecting 10% samples of the entire training; the initial unlabeled pool is the rest 90%. Then 10 AL rounds are performed; in each round, an unlabeled batch of 4% samples of the entire training set is queried, so the total annotation budget is 50% of the training set. We use the BACE, BBBP, HIV and SIDER datasets from the widely-used

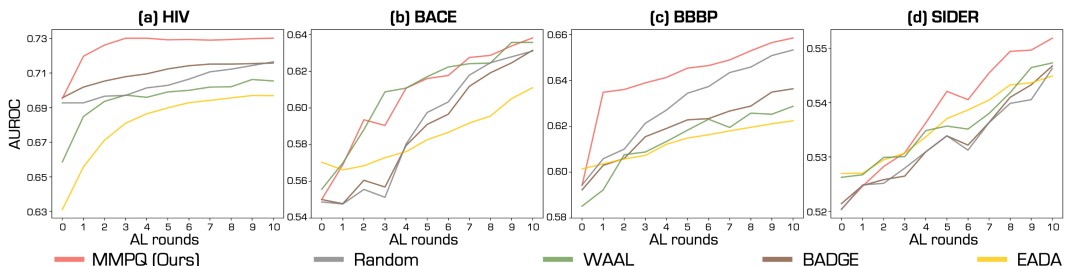

**Figure 2:** Active learning performance of MMPQ (ours) and baseline hybrid methods. "Round 0" corresponds to the performance on initial labeled pool. Note y-axis has an offset for clear comparison.

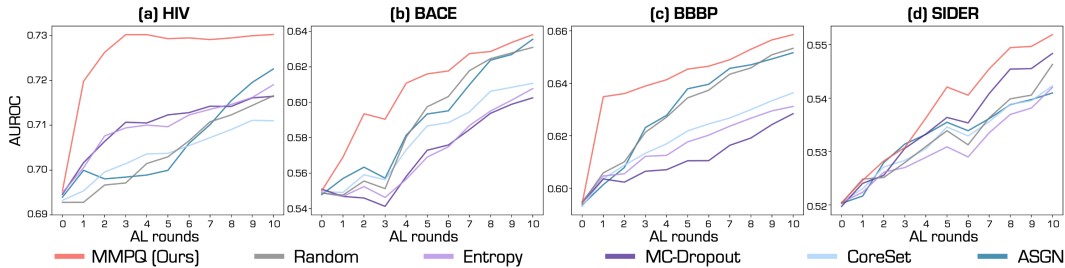

**Figure 3:** Active learning performance of MMPQ (ours) and uncertainty-based or diversity-based methods. "Round 0" corresponds to the performance on initial labeled pool. Note y-axis has an offset for clear comparison.

MoleculeNet benchmark [5] (also included in the Open Graph Benchmark [46]). Statistics and detailed descriptions of the datasets are in Appx. A.5. Following [4], we use *scaffold split*, with train/val/test = 80%/10%/10%. We use AUROC as the performance metric, as suggested in [5]. Please refer to Appx. A.6 for implementation details.

## 5.2 Active Learning Performance

We compare MMPQ against following 8 baselines, with **U**, **D** and **H** denoting **U**ncertainty-based, **D**iversity-based and **H**ybrid methods respectively: **Random** (random selection), **Entropy (U)** (selecting samples with the largest prediction entropy), **MC-Dropout (U)** [8], **CoreSet (D)** [9], **ASGN (D)** [47], **BADGE (H)** [10], **WAAL (H)** [12], **EADA (H)** [13]. Among them, ASGN is the only existing method that investigates AL in molecular property prediction. BADGE, WAAL and EADA are representative hybrid methods, and are state-of-the-art on many image classification datasets. We note that there are other hybrid methods [11, 14, 15], but their code is not or only partly released. We fail to reproduce their results, and hence do not include them for comparison. To avoid cluttered presentation, we show results of baseline hybrid methods in Fig. 2, and those of uncertainty-based or diversity-based methods in Fig. 3. In both figures, we include results of our MMPQ and the Random baseline.

**Results:** From the figures, we can see that our MMPQ outperforms the baselines on all 4 datasets. Specifically, on HIV, our MMPQ achieves 0.7302 AUROC in the 3-rd active round (using only 22% annotations of the entire training set), which is very close to the performance of using 100% of the annotations (0.7344). This also explains why the performance of MMPQ almost saturates after the 3-rd round. Furthermore, the performance of hybrid methods requiring trade-off hyperparameters, *i.e.*, WAAL and EADA, is not stable. In particular, though WAAL achieves performance comparable to our proposed MMPQ on BACE, it performs quite unsatisfactorily on HIV. Similarly, EADA performs well on SIDER but is the worst baseline on HIV. By contrast, our method achieves consistently superior performance. One may note that the performance of WAAL and EADA at round 0 is quite different from that of other methods. The reason is that, in other methods, the task learner is only trained with classification loss on the currently labeled data. Contrarily, in WAAL and EADA, apart from classification loss, the task learner is also trained with some auxiliary loss (*i.e.*, adversarial loss

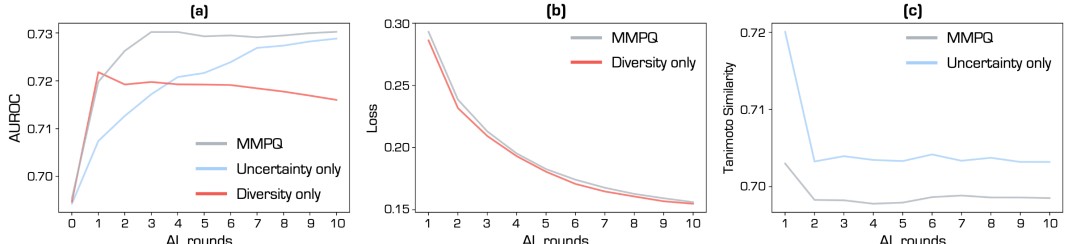

**Figure 4: (a)** AL performance of MMPQ, U.O. and D.O. strategies. **(b)** Mean ground-truth-label loss of data queried by MMPQ or D.O. strategy. **(c)** Mean average Tanimoto similarity of data queried by MMPQ or U.O. strategy.

### 5.3 Ablation Studies

Here we conduct ablative experiments on HIV, which is the *largest* dataset used (see Tab. 1).

#### 5.3.1 Uncertainty or Diversity Only

In Sec. 4.1.1, we show that our MMPQ strategy captures both uncertainty and diversity through the two terms in Eqn. (5) respectively. Here we ablatively study the effectiveness of the two terms. Since we have only 1 target property on HIV, we use $\hat{y}$ instead of $\hat{\mathbf{y}}$. Specifically, under the setup described in Sec. 5.1, we compare our MMPQ with another two strategies based on the trained EBM:

- The **U.O.** strategy that considers **U**ncertainty **O**nly: querying data with minimum $p^M = \max_{\hat{y}} p(\hat{y}|G)$. Let $\hat{y}^*$ denote the predicted label that achieves $p^M$. Then, based on the learned EBM, $p^M$ can be calculated by:

$$p^M = \frac{\exp(-E(G, \hat{y}^*))}{\sum_{\hat{y} \in \{0,1\}} \exp(-E(G, \hat{y}))}. \tag{12}$$

- The **D.O.** strategy that considers **D**iversity **O**nly: querying those with minimum $p_L(G)$:

$$p_L(G) \propto \sum_{\hat{y}\{0,1\}} \exp(-E(G, \hat{y})). \tag{13}$$

In Fig. 4 (a), as AL proceeds, the performance of the U.O. strategy rises slower than that of MMPQ or the D.O. strategy, though the final performance of U.O. (at round 10) is as good as MMPQ. On the other hand, the D.O. strategy reaches a peak very quickly (*i.e.*, at the 2-nd AL round), but then its performance degrades as more data are annotated for training. One possible reason for such degradation is that the data queried in later rounds cannot provide the task learner with useful information about the learning task. Adding these data to the training pool may lead to overfitting, since more data means more training iterations. This shows that uncertainty and diversity are complementary to each other — diversity is important in early AL stages, while uncertainty is critical in later stages. Interestingly, this corroborates the finding in [48].

Furthermore, we then dig deeper into the advantages of the MMPQ strategy, by examining how the two terms in Eqn. (5) affect the queried data. For studying the effectiveness of the uncertainty-based term $\max_{\hat{\mathbf{y}}} p(\hat{\mathbf{y}}|G)$, we examine whether the queried data of the MMPQ strategy have higher uncertainty than those of the D.O. strategy (since MMPQ and D.O. only differ in this term). Note that, since HIV has only 1 property of interest, this term becomes $p(\hat{y}|G)$. For measuring uncertainty, distribution-based metrics such as entropy and classification margin are often used. However, for binary classification (as in our case), these metrics are equivalent to selecting those with the smallest $p(\hat{y}|G)$. Therefore, instead of these metrics, we adopt the ground-truth-label loss, as used in [30]. For the $t$-th AL round, we calculate the mean loss of data in the current queries $\mathcal{D}_L^t$. As shown in Fig. 4 (b), compared with the D.O. strategy, queries of our MMPQ have larger ground-truth-label loss, suggesting larger uncertainty of the task learner.

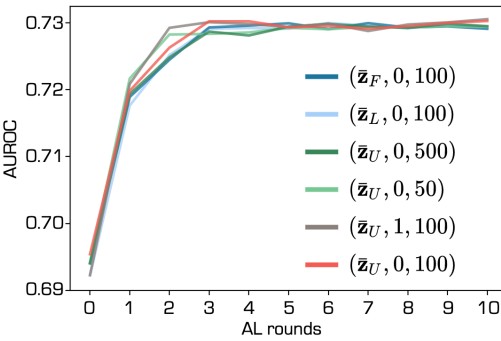 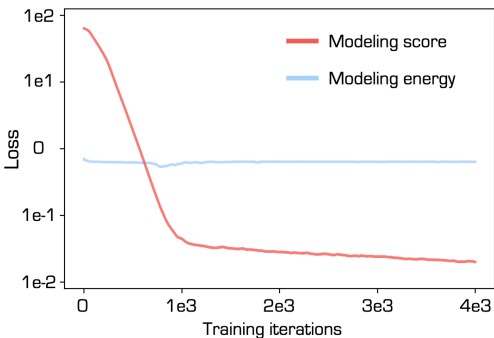

**Figure 5:** Performance of different $(\mathbf{z}_0, \hat{y}_0, K)$. **Figure 6:** DSM loss of modeling score or energy.

For the diversity-based term $p_L(G)$, we examine whether queries of MMPQ have smaller chemical similarity (*i.e.*, larger diversity) than those of U.O. strategy (since of MMPQ and U.O. only differ in this term). We adopt the Tanimoto similarity [49], which is a widely used expert-defined molecular similarity metric. Formally, let $T_{ij}$ denote the Tanimoto similarity between molecule $G_i$ and $G_j$, and we calculate the mean Average Similarity (mAS) among molecules in $\mathcal{D}_L^t$:

$$mAS = \frac{1}{N_L^t(N_L^t - 1)} \sum\nolimits_{G_i, G_j \in \mathcal{D}_L^t, j \neq i} T_{ij}. \tag{14}$$

Fig. 4 (c) shows that queries of MMPQ are less chemically similar than those of U.O., implying larger diversity.

### 5.3.2 Robustness of Energy Calculation

In this part, we investigate the robustness of MMPQ w.r.t. how energy is calculated. Specifically, we consider the choice of zero-energy point, and the number of points for approximating the integral (*i.e.*, $K$ in Eqn. (18)).

In the above experiments, we set the zero-energy point as $(\mathbf{z}_0 = \bar{\mathbf{z}}_U, \hat{y}_0 = 0)$, where $\bar{\mathbf{z}}_U$ is the where $\bar{\mathbf{z}}_U$ is the mean embedding of the unlabeled pool, and let $K = 100$. We name this setting the "default setting", and denote it with the triple $(\bar{\mathbf{z}}_U, 0, 100)$.

Then, based on $(\bar{\mathbf{z}}_U, 0, 100)$, we vary one of the three hyperparameters while keeping the other two unchanged, and run AL experiments under the setup described in Sec. 5.1. Specifically, we set (1) $\mathbf{z}_0 \in \{\bar{\mathbf{z}}_L, \bar{\mathbf{z}}_F\}$, where $\bar{\mathbf{z}}_L$ and $\bar{\mathbf{z}}_F$ are the mean embedding of the **L**abeled pool and that of the **F**ull training set; (2) $\hat{y}_0 = 1$; (3) $K \in \{50, 500\}$. Fig. 5 shows the AL performance of the above settings and the default one. We can see that the AL performance of different settings are similar, which demonstrates that the MMPQ strategy is robust to the above hyperparameters.

### 5.3.3 Implementing EBM by Modeling Energy

As introduced in Sec. 4.2, we instantiate the EBM as a score net learning the score of the true distribution. An alternative is to implement the EBM as an energy net modeling the energy function.

We try this alternative in our experiments, but find that the training process fails to converge. Specifically, the energy net is also built on the embedding space of the AE, and has the same architecture as the score net, except that the final layer has an output dimension of 1 and a ReLU activation (since the energy is a non-negative scalar). The training objective of the energy net (denoted as $E_\theta$) is

$$\mathbb{E}_{\substack{\mathbf{x} \in \mathcal{D}_L, \\ \tilde{\mathbf{x}} \sim \mathcal{N}(\tilde{\mathbf{x}}|\mathbf{x}, \sigma)}} \left[ \left\| -\nabla_{\mathbf{x}} E_\theta(\tilde{\mathbf{x}}) + \frac{\tilde{\mathbf{x}} - \mathbf{x}}{\sigma^2} \right\|_2^2 \right]. \tag{15}$$

Fig. 6 shows the loss curve (in a log scale) under best-tuned hyperparameters (*i.e.*, those yielding lowest loss). For comparison, the loss curve on HIV of our score net used in the MMPQ experiment in Sec. 5.2 is also given. We can see that, even with the best-tuned hyperparameters, the training process of $E_\theta$ cannot converge well.

## 6 Conclusion and Limitation

We propose Maximum Minimum Probability Querying (MMPQ), a hybrid active learning method for molecular property prediction, without the need to manually trade off between uncertainty and diversity. The strategy is based on an EBM that models the joint distribution of labeled data and task learner's prediction. The EBM is built in an embedding space learned by an auto-encoder that reconstructs molecules' SMILES string. We propose training the EBM via denoising score matching. Once the EBM is trained, MMPQ selects data according to one single selection criterion that naturally captures the uncertainty of the task learner and high diversity w.r.t. in the data space.

One limitation of our approach is that it is only applicable to binary classification tasks. This is because we use the maximum prediction probability as an uncertainty metric, which is equivalent to the commonly-used entropy metric only in the binary case. When extending to categorical classification or regression, our used metric may not well reflect model uncertainty. In Appx. A.8, we discuss one possible solution to address this limitation by utilizing a new uncertainty metric. In our further work, we look forward to introducing our proposed model to model zoos for knowledge reassembly [50, 51].

## Acknowledgement

This research is supported by the National Research Foundation Singapore under its AI Singapore Programme (Award Number: AISG2-RP-2021-023). Tingyang Xu and Xinchao Wang are the co-corresponding authors.

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

# A    Appendix

## A.1    Pool-based batch-mode active learning setting

In this setting, we are given an initial pool of labeled data $\mathcal{D}_L^0$ of size $N_L^0$, and an unlabeled pool $\mathcal{D}_U^0$ of size $N_U^0 > N_L^0$. Our goal is to design an AL algorithm that performs $T$ rounds of querying [7]. In the $t$-th round $(1 \leqslant t \leqslant T)$, a batch of $b$ samples, denoted as $\mathcal{B}^t$, are selected from $\mathcal{D}_U^{t-1}$. Then, an oracle (*e.g.*, in our case, a chemist) annotates the queried samples, which are then moved from the unlabeled pool to the labeled pool. Formally, let $\mathcal{B}_{\text{anno}}^t$ denote annotated selected batch, and then the pools are updated by $\mathcal{D}_L^t = \mathcal{D}_L^{t-1} \cup \mathcal{B}_{\text{anno}}^t$, and $\mathcal{D}_U^t = \mathcal{D}_U^{t-1} \setminus \mathcal{B}^t$; accordingly $N_L^t = N_L^{t-1} + b$, $N_U^t = N_U^{t-1} - b$. The obtained $\mathcal{D}_L^t$ is then used to train a *task learner* $h(\cdot)$ — the model that performs the target learning task at hand (*e.g.*, molecular property prediction in our case). In this paper, the task learner is a Graph Neural Networks (GNN) [52].

Note that, the union of $\mathcal{D}_L^t$ and $\mathcal{D}_U^t$ is always the whole training set, *i.e.*, $\mathcal{D}_{\text{train}} = \mathcal{D}_L^t \cup \mathcal{D}_U^t, \forall t \geqslant 0$. Aside from $\mathcal{D}_{\text{train}}$, we also have a validation set $\mathcal{D}_{\text{val}}$ and a test set $\mathcal{D}_{\text{test}}$, which are held-out and disjoint from $\mathcal{D}_{\text{train}}$, for performing model selection and evaluation on the task learner. For brevity, we omit the round index $t$ unless necessary.

## A.2    Calculating minimum energy for large $n$

As mentioned in Sec. 4.1.2, we can calculate $\min_{\hat{\mathbf{y}}} E(G, \hat{\mathbf{y}})$ in a task-wise manner, under the assumption that the $n$ labels are conditionally independent. Here we elaborate on the calculation.

In multi-label classification, the $n$ labels are often assumed to be independent given the input. In our case, this is formulated as: $p(\hat{\mathbf{y}}|G) = \prod_{i=1}^n p(y_i|G)$. Thus, we have

$$
\begin{aligned}
& \arg\min_{\hat{\mathbf{y}}} E(G, \hat{\mathbf{y}}) \\
&= \arg\max_{\hat{\mathbf{y}}} q(G, \hat{\mathbf{y}}) \\
&= \arg\max_{\hat{\mathbf{y}}} p(\hat{y}_1, \cdots, \hat{y}_n|G) p_L(G) \\
&= \arg\max_{\hat{\mathbf{y}}} \left( \prod_{i=1}^n p(\hat{y}_i|G) \right) \\
&= \left( \arg\max_{\hat{y}_1} p(\hat{y}_1|G), \cdots, \arg\max_{\hat{y}_n} p(\hat{y}_n|G) \right).
\end{aligned}
\tag{16}
$$

This shows that we can calculate $\arg\min_{\hat{\mathbf{y}}} E(G, \hat{\mathbf{y}})$ by simply taking $\left( \arg\max_{\hat{y}_1} p(\hat{y}_1|G), \cdots, \arg\max_{\hat{y}_n} p(\hat{y}_n|G) \right)$, without the need of calculating the energy for all $2^n$ possible combinations of $(\hat{y}_1, \cdots, \hat{y}_n)$.

## A.3    Energy calculation

Our EBM models the score $\nabla_{\mathbf{x}} \log q(\mathbf{x})$ instead of the energy $E(\mathbf{x})$. For obtaining the energy value $E(\mathbf{x})$, we use summation to approximate integral over $\nabla_{\mathbf{x}} \log q(\mathbf{x})$ after training the EBM.

Specifically, the energy of any point $\mathbf{x}$ can be calculated through the following line integral:

$$
E(\mathbf{x}) = E(\mathbf{x}_0) + \int_P \nabla_{\mathbf{x}} \log q(\mathbf{x}) \cdot d\mathbf{x},
\tag{17}
$$

where $\cdot$ denotes a vector inner product. The two terms on the right hand side are explained as follows.

The first term $E(\mathbf{x}_0)$ is the energy of an arbitrarily chosen reference point $\mathbf{x}_0$. For selecting queries, we only need the relative energy value. Therefore, without loss of generality, we can take any reference point $\mathbf{x}_0$ as the zero-energy point, *i.e.*, letting $E(\mathbf{x}_0) = 0$.

The second term is an integral along a path $P$ from $\mathbf{x}_0$ to $\mathbf{x}$. Since the true score $\nabla_{\mathbf{x}} \log q(\mathbf{x})$ is a conservative vector field, the integral result does not depend on the choice of $P$ (assuming that the EBM approximates $\nabla_{\mathbf{x}} \log q(\mathbf{x})$ well). We thus calculate the integral along the directed line segment from $\mathbf{x}_0$ to $\mathbf{x}$, denoted as $\int_{\mathbf{x}_0}^{\mathbf{x}}$.

Finally, the (relative) energy of $\mathbf{x}$ can be calculated by

$$
\begin{aligned}
E(\mathbf{x}) &= \int_{\mathbf{x}_0}^{\mathbf{x}} \nabla_{\mathbf{x}} \log q(\mathbf{x}) \cdot \mathrm{d}\mathbf{x} \\
&\approx \sum_{k=0}^{K} (\mathbf{x}_{k+1} - \mathbf{x}_k) \cdot \nabla_{\mathbf{x}} \log q(\mathbf{x}_{k+1}) \\
&= \sum_{k=0}^{K} (\mathbf{x}_{k+1} - \mathbf{x}_k) \cdot s_{\theta}(\mathbf{x}_{k+1}),
\end{aligned}
\tag{18}
$$

where $\{\mathbf{x}_1, \cdots, \mathbf{x}_K\}$ denote $K$ points evenly distributed along the directed line segment, and $\mathbf{x}_{K+1} \triangleq \mathbf{x}$.

### A.4 Pseudo codes

Alg. 1 and Alg. 2 show the pseudo codes of MMPQ and the model training process, respectively. Notations used can be found in Appx. A.1.

---

**Algorithm 1** Active learning with MMPQ

---

**Input:** Initial labeled pool $\mathcal{D}_L^0$, unlabeled pool $\mathcal{D}_U^0$, number of rounds $T$, number of queries per round $b$, EBM, task learner
 1: Train task learner with $\mathcal{D}_L^0$, perform model selection using $\mathcal{D}_{\mathrm{val}}$, and test the learner on $\mathcal{D}_{\mathrm{test}}$
 2: **for** $t \in \{0, \cdots, T\}$ **do**
 3:    Train the EBM (Sec. 4.3)
 4:    // perform active selection
 5:    Select a batch of $b$ samples $\mathcal{B}^t$ according to Eqn. (4)
 6:    Annotate $\mathcal{B}^t$ and obtain $\mathcal{B}_{\mathrm{anno}}^t$
 7:    $\mathcal{D}_L^t = \mathcal{D}_L^{t-1} \cup \mathcal{B}_{\mathrm{anno}}^t, \mathcal{D}_U^t = \mathcal{D}_U^{t-1} \setminus \mathcal{B}^t$
 8:    // train and test task learner
 9:    Train task learner with $\mathcal{D}_L^t$, perform model selection using $\mathcal{D}_{\mathrm{val}}$, and test the learner on $\mathcal{D}_{\mathrm{test}}$
10: **end for**
**Output:** Performance on $\mathcal{D}_{\mathrm{test}}$ for $t \in \{0, \cdots, T\}$

---

**Algorithm 2** Training process

---

**Input:** EBM $s_{\theta}$, AE encoder Enc and decoder Dec, and task learner $h$ (trained and fixed)
 1: **while** not converge **do**
 2:    Randomly sample $G$ from $\mathcal{D}_L$, $G'$ from $\mathcal{D}_U$
 3:    Sample $\hat{\mathbf{y}}$ by drawing $\hat{y}_i$ from $\mathrm{Ber}(h(G)_i)$
 4:    $\mathbf{x} = (\mathrm{Enc}(\mathrm{Sml}(G)), \hat{\mathbf{y}})$
 5:    Calculate $L_{\mathrm{joint}}$ with Eqn. (9), (10), (11)
 6:    Update $s_{\theta}$, Enc and Dec with $L_{\mathrm{joint}}$
 7: **end while**
**Output:** Trained EBM and AE

---

### A.5 Dataset information

We use the BACE, BBBP, HIV and SIDER datasets from MoleculeNet [5]. Here we give a brief introduction to these datasets:

- **BACE**: Human $\beta$-secretase 1 (a.k.a. BACE-1) is an enzyme in human body. It is recently found that inhibition of BACE-1 can slow down the development of Alzheimer's disease. The BACE dataset contains experimentally measured binding (*i.e.*, effective in inhibition) results (binarized) of 1,513 candidate inhibitors of BACE-1.

- **BBBP**: The Blood–Brain Barrier (BBB) is a highly selective semipermeable border that prevents solutes in the blood from non-selectively crossing into the extracellular fluid of the central nervous system. For designing drugs to cure some brain disorder, one major challenge is to ensure that the obtained drug is able to go through BBB. The BBBP dataset provides binary labels for 2,039 molecules on their ability to permeate BBB.

- **HIV**: The HIV dataset provides the experimentally tested ability to inhibit HIV replication for 41,127 molecules.
- **SIDER**: SIDER is an abbreviation of Side Effect Resource, which is a database of marketed drugs and their adverse drug reactions (*i.e.*, side effects). The SIDER dataset contains the results of 1,427 drugs on 27 kinds of side effects.

Statistics of the datasets are in Tab. 1.

**Table 1:** Statistics of used datasets

|  | BACE | SIDER | BBBP | HIV |
|---|---|---|---|---|
| Number of Tasks | 1 | 27 | 1 | 1 |
| Number of Molecules | 1,513 | 1,427 | 2,039 | 41,127 |

We note that the TOX21, TOXCAST, MUV and PCBA datasets in the MoleculeNet benchmark [5] are also often used. However, we find they contain a large number of molecules whose properties not fully provided by the dataset creator.

To see this, for each property of the 4 datasets, we show in Fig. 7 the numbers of molecules whose ground truth labels are provided or not. We can see that, there are a large number of label-not-provided molecules for each property that needs to be predicted. This violates the general assumption in AL that the oracle can correctly annotate each query. Thus we do not run experiments on these datasets.

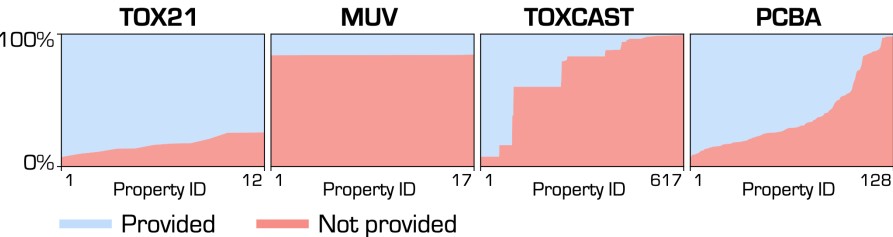

**Figure 7:** Ratios of molecules whose ground truth properties are provided or not. The x-axis ticks correspond to the index of properties. We sort the properties by the ratio of label-not-provided molecules in ascending order.

### A.6 Implementation details

To obtain reliable results, we randomly sample 50 initial labeled pools in advance, and run each AL experiment with all these pools and 10 random seeds per pool. The reported performance is averaged across all initial pools and seeds (*i.e.*, $50 \times 10 = 500$ runs).

The AE for reconstructing the SMILES strings is a Transformer [22, 53], which has been shown to be effective in sequence modeling. The EBM is a 5-layer Multi-Layer Perceptron with residual connections, ReLU activation function [54], and Layer Normalization [55] between each hidden layer. We use the RdKit library[3] to pre-generate SMILES strings, and follow [22] to tokenize the strings. We instantiate the task learner with a 5-layer GINE architecture [4], which is widely used for molecular property prediction [4, 36, 37].

Before the active learning procedure, the AE is pretrained on the dataset of interest for 5,000 epochs with learning rate of 5e-4. In each AL round, the auto-encoder and EBM are jointly trained with batch size of 128. The learning rates are 2e-4 for the auto-encoder and 1e-3 for the EBM. The standard deviation of the Gaussian noise used in DSM is 10. The criterion for training convergence is that there are 10 batches whose DSM loss (see Eqn. (10)) is no larger than 0.015, which is obtained in our pilot experiments. The maximum number of training epochs is 2500. The used optimizer is Adam. The task learner is trained for 50 epochs with batch size of 128, learning rate of 1e-3, and the Adam optimizer. Note that the task learner is re-initialized before it is trained and tested. For calculating the

---

[3]https://www.rdkit.org/

(relative) energy, we use points for approximation, *i.e.*, $K = 100$ (see Eqn. (18)), and set zero-energy point as $(\mathbf{z}_0 = \bar{\mathbf{z}}_U, \hat{y}_0 = 0)$, where $\bar{\mathbf{z}}_U$ is the mean embedding of the unlabeled pool:

$$\bar{\mathbf{z}}_U = \frac{1}{N_U} \sum\nolimits_{G \in \mathcal{D}_U} \text{Enc}(\text{Sml}(G)). \qquad (19)$$

### A.7 Numerical results

Tab. 2, Tab. 3, Tab. 4 and Tab. 5 show the numerical results (including mean and standard deviation) of Fig. 2 and Fig. 3. Tab. 6 shows the numerical results of Fig. 4. Tab. 7 shows the numerical results of Fig. 5.

**Table 2:** Numerical results on BACE. "Round 0" corresponds to performance on initial labeled pools.

| | 0 | 1 | 2 | 3 | 4 | 5 |
|---|---|---|---|---|---|---|
| Random | $0.5488 \pm .0090$ | $0.5477 \pm .0233$ | $0.5556 \pm .0243$ | $0.5513 \pm .0204$ | $0.5803 \pm .0202$ | $0.5975 \pm .0196$ |
| Entropy | $0.5489 \pm .0122$ | $0.5469 \pm .0172$ | $0.5525 \pm .0166$ | $0.5463 \pm .0153$ | $0.5566 \pm .0161$ | $0.5691 \pm .0152$ |
| MC-Dropout | $0.5510 \pm .0110$ | $0.5469 \pm .0158$ | $0.5460 \pm .0172$ | $0.5413 \pm .0163$ | $0.5581 \pm .0142$ | $0.5730 \pm .0144$ |
| CoreSet | $0.5497 \pm .0113$ | $0.5491 \pm .0204$ | $0.5591 \pm .0225$ | $0.5563 \pm .0212$ | $0.5731 \pm .0216$ | $0.5867 \pm .0198$ |
| ASGN | $0.5480 \pm .0120$ | $0.5570 \pm .0234$ | $0.5634 \pm .0217$ | $0.5572 \pm .0220$ | $0.5814 \pm .0209$ | $0.5934 \pm .0219$ |
| WAAL | $0.5557 \pm .0076$ | $0.5699 \pm .0141$ | $0.5882 \pm .0136$ | $0.6087 \pm .0127$ | $0.6108 \pm .0155$ | $0.6170 \pm .0148$ |
| EADA | $0.5704 \pm .0085$ | $0.5662 \pm .0126$ | $0.5685 \pm .0101$ | $0.5729 \pm .0118$ | $0.5762 \pm .0104$ | $0.5827 \pm .0099$ |
| Badge | $0.5501 \pm .0118$ | $0.5478 \pm .0168$ | $0.5606 \pm .0136$ | $0.5569 \pm .0157$ | $0.5795 \pm .0156$ | $0.5910 \pm .0143$ |
| MMPQ (Ours) | $0.5501 \pm .0129$ | $0.5690 \pm .0175$ | $0.5935 \pm .0183$ | $0.5904 \pm .0172$ | $0.6108 \pm .0164$ | $0.6160 \pm .0152$ |

| | 6 | 7 | 8 | 9 | 10 | |
|---|---|---|---|---|---|---|
| Random | $0.6032 \pm .0201$ | $0.6178 \pm .0187$ | $0.6246 \pm .0190$ | $0.6278 \pm .0192$ | $0.6309 \pm .0188$ | |
| Entropy | $0.5749 \pm .0149$ | $0.5869 \pm .0158$ | $0.5958 \pm .0163$ | $0.6012 \pm .0142$ | $0.6077 \pm .0156$ | |
| MC-Dropout | $0.5759 \pm .0139$ | $0.5846 \pm .0132$ | $0.5938 \pm .0153$ | $0.5989 \pm .0137$ | $0.6025 \pm .0139$ | |
| CoreSet | $0.5885 \pm .0186$ | $0.5946 \pm .0200$ | $0.6062 \pm .0189$ | $0.6084 \pm .0176$ | $0.6107 \pm .0191$ | |
| ASGN | $0.5952 \pm .0175$ | $0.6099 \pm .0192$ | $0.6237 \pm .0208$ | $0.6269 \pm .0201$ | $0.6355 \pm .0188$ | |
| WAAL | $0.6223 \pm .0147$ | $0.6241 \pm .0132$ | $0.6244 \pm .0130$ | $0.6356 \pm .0137$ | $0.6357 \pm .0118$ | |
| EADA | $0.5867 \pm .0127$ | $0.5918 \pm .0101$ | $0.5955 \pm .0102$ | $0.6050 \pm .0087$ | $0.6110 \pm .0122$ | |
| Badge | $0.5965 \pm .0117$ | $0.6117 \pm .0163$ | $0.6191 \pm .0151$ | $0.6245 \pm .0161$ | $0.6314 \pm .0144$ | |
| MMPQ (Ours) | $0.6176 \pm .0133$ | $0.6274 \pm .0165$ | $0.6286 \pm .0143$ | $0.6337 \pm .0146$ | $0.6381 \pm .0142$ | |

**Table 3:** Numerical results on BBBP. "Round 0" corresponds to performance on initial labeled pools.

| | 0 | 1 | 2 | 3 | 4 | 5 |
|---|---|---|---|---|---|---|
| Random | $0.5944 \pm .0099$ | $0.6059 \pm .0212$ | $0.6101 \pm .0218$ | $0.6214 \pm .0213$ | $0.6271 \pm .0216$ | $0.6344 \pm .0183$ |
| Entropy | $0.5950 \pm .0085$ | $0.6047 \pm .0122$ | $0.6055 \pm .0156$ | $0.6122 \pm .0157$ | $0.6125 \pm .0139$ | $0.6176 \pm .0116$ |
| MC-Dropout | $0.5945 \pm .0088$ | $0.6035 \pm .0130$ | $0.6023 \pm .0172$ | $0.6065 \pm .0144$ | $0.6071 \pm .0153$ | $0.6105 \pm .0147$ |
| CoreSet | $0.5931 \pm .0087$ | $0.6036 \pm .0199$ | $0.6088 \pm .0195$ | $0.6134 \pm .0163$ | $0.6169 \pm .0182$ | $0.6218 \pm .0149$ |
| ASGN | $0.5936 \pm .0074$ | $0.6013 \pm .0176$ | $0.6079 \pm .0184$ | $0.6231 \pm .0161$ | $0.6278 \pm .0158$ | $0.6379 \pm .0145$ |
| WAAL | $0.5852 \pm .0102$ | $0.5923 \pm .0177$ | $0.6076 \pm .0169$ | $0.6088 \pm .0173$ | $0.6134 \pm .0159$ | $0.6183 \pm .0164$ |
| EADA | $0.6014 \pm .0075$ | $0.6036 \pm .0112$ | $0.6058 \pm .0137$ | $0.6073 \pm .0130$ | $0.6122 \pm .0136$ | $0.6149 \pm .0087$ |
| Badge | $0.5923 \pm .0068$ | $0.6030 \pm .0086$ | $0.6060 \pm .0104$ | $0.6155 \pm .0075$ | $0.6191 \pm .0092$ | $0.6229 \pm .0075$ |
| MMPQ (Ours) | $0.5943 \pm .0075$ | $0.6349 \pm .0121$ | $0.6361 \pm .0123$ | $0.6389 \pm .0126$ | $0.6413 \pm .0097$ | $0.6454 \pm .0087$ |

| | 6 | 7 | 8 | 9 | 10 | |
|---|---|---|---|---|---|---|
| Random | $0.6373 \pm .0198$ | $0.6434 \pm .0198$ | $0.6458 \pm .0185$ | $0.6509 \pm .0190$ | $0.6533 \pm .0184$ | |
| Entropy | $0.6202 \pm .0123$ | $0.6237 \pm .0135$ | $0.6267 \pm .0105$ | $0.6295 \pm .0127$ | $0.6312 \pm .0125$ | |
| MC-Dropout | $0.6105 \pm .0126$ | $0.6164 \pm .0133$ | $0.6191 \pm .0143$ | $0.6243 \pm .0122$ | $0.6284 \pm .0118$ | |
| CoreSet | $0.6245 \pm .0153$ | $0.6268 \pm .0129$ | $0.6300 \pm .0151$ | $0.6334 \pm .0170$ | $0.6363 \pm .0166$ | |
| ASGN | $0.6396 \pm .0142$ | $0.6457 \pm .0141$ | $0.6470 \pm .0145$ | $0.6493 \pm .0149$ | $0.6516 \pm .0152$ | |
| WAAL | $0.6231 \pm .0144$ | $0.6195 \pm .0139$ | $0.6257 \pm .0171$ | $0.6253 \pm .0168$ | $0.6287 \pm .0139$ | |
| EADA | $0.6163 \pm .0103$ | $0.6180 \pm .0115$ | $0.6196 \pm .0091$ | $0.6211 \pm .0095$ | $0.6224 \pm .0104$ | |
| Badge | $0.6234 \pm .0091$ | $0.6267 \pm .0079$ | $0.6288 \pm .0094$ | $0.6350 \pm .0077$ | $0.6364 \pm .0076$ | |
| MMPQ (Ours) | $0.6465 \pm .0083$ | $0.6490 \pm .0087$ | $0.6530 \pm .0112$ | $0.6565 \pm .0084$ | $0.6585 \pm .0082$ | |

**Table 4:** Numerical results on SIDER. "Round 0" corresponds to performance on initial labeled pools.

|  | 0 | 1 | 2 | 3 | 4 | 5 |
|---|---|---|---|---|---|---|
| Random | $0.5203 \pm .0072$ | $0.5248 \pm .0165$ | $0.5251 \pm .0172$ | $0.5279 \pm .0150$ | $0.5309 \pm .0155$ | $0.5339 \pm .0138$ |
| Entropy | $0.5205 \pm .0081$ | $0.5224 \pm .0123$ | $0.5261 \pm .0168$ | $0.5270 \pm .0103$ | $0.5289 \pm .0119$ | $0.5309 \pm .0105$ |
| MC-Dropout | $0.5197 \pm .0084$ | $0.5240 \pm .0146$ | $0.5255 \pm .0144$ | $0.5306 \pm .0110$ | $0.5333 \pm .0115$ | $0.5364 \pm .0124$ |
| CoreSet | $0.5205 \pm .0073$ | $0.5230 \pm .0161$ | $0.5271 \pm .0184$ | $0.5283 \pm .0167$ | $0.5305 \pm .0161$ | $0.5346 \pm .0145$ |
| ASGN | $0.5203 \pm .0073$ | $0.5217 \pm .0163$ | $0.5279 \pm .0159$ | $0.5314 \pm .0161$ | $0.5332 \pm .0148$ | $0.5354 \pm .0159$ |
| WAAL | $0.5262 \pm .0064$ | $0.5267 \pm .0150$ | $0.5299 \pm .0159$ | $0.5300 \pm .0144$ | $0.5348 \pm .0137$ | $0.5357 \pm .0144$ |
| EADA | $0.5269 \pm .0069$ | $0.5270 \pm .0091$ | $0.5295 \pm .0100$ | $0.5307 \pm .0085$ | $0.5337 \pm .0104$ | $0.5371 \pm .0058$ |
| Badge | $0.5213 \pm .0067$ | $0.5247 \pm .0075$ | $0.5258 \pm .0100$ | $0.5265 \pm .0084$ | $0.5310 \pm .0068$ | $0.5339 \pm .0080$ |
| MMPQ (Ours) | $0.5204 \pm .0054$ | $0.5246 \pm .0087$ | $0.5282 \pm .0070$ | $0.5308 \pm .0087$ | $0.5363 \pm .0068$ | $0.5421 \pm .0068$ |

|  | 6 | 7 | 8 | 9 | 10 |
|---|---|---|---|---|---|
| Random | $0.5313 \pm .0134$ | $0.5363 \pm .0137$ | $0.5399 \pm .0150$ | $0.5405 \pm .0148$ | $0.5463 \pm .0164$ |
| Entropy | $0.5290 \pm .0124$ | $0.5335 \pm .0118$ | $0.5369 \pm .0107$ | $0.5381 \pm .0118$ | $0.5420 \pm .0094$ |
| MC-Dropout | $0.5356 \pm .0145$ | $0.5408 \pm .0124$ | $0.5454 \pm .0085$ | $0.5455 \pm .0105$ | $0.5483 \pm .0103$ |
| CoreSet | $0.5330 \pm .0155$ | $0.5356 \pm .0160$ | $0.5388 \pm .0148$ | $0.5395 \pm .0169$ | $0.5423 \pm .0146$ |
| ASGN | $0.5339 \pm .0136$ | $0.5361 \pm .0141$ | $0.5388 \pm .0140$ | $0.5397 \pm .0130$ | $0.5410 \pm .0125$ |
| WAAL | $0.5351 \pm .0130$ | $0.5380 \pm .0147$ | $0.5418 \pm .0130$ | $0.5464 \pm .0130$ | $0.5473 \pm .0121$ |
| EADA | $0.5387 \pm .0081$ | $0.5405 \pm .0080$ | $0.5433 \pm .0081$ | $0.5436 \pm .0090$ | $0.5449 \pm .0085$ |
| Badge | $0.5322 \pm .0084$ | $0.5363 \pm .0080$ | $0.5410 \pm .0081$ | $0.5433 \pm .0075$ | $0.5468 \pm .0070$ |
| MMPQ (Ours) | $0.5405 \pm .0066$ | $0.5455 \pm .0079$ | $0.5494 \pm .0066$ | $0.5497 \pm .0073$ | $0.5519 \pm .0065$ |

**Table 5:** Numerical results on HIV. "Round 0" corresponds to performance on initial labeled pools.

|  | 0 | 1 | 2 | 3 | 4 | 5 |
|---|---|---|---|---|---|---|
| Random | $0.6928 \pm .0109$ | $0.6928 \pm .0241$ | $0.6967 \pm .0260$ | $0.6971 \pm .0224$ | $0.7015 \pm .0244$ | $0.7030 \pm .0213$ |
| Entropy | $0.6949 \pm .0098$ | $0.7005 \pm .0199$ | $0.7076 \pm .0213$ | $0.7094 \pm .0158$ | $0.7100 \pm .0139$ | $0.7097 \pm .0164$ |
| MC-Dropout | $0.6946 \pm .0075$ | $0.7016 \pm .0118$ | $0.7063 \pm .0122$ | $0.7106 \pm .0140$ | $0.7105 \pm .0115$ | $0.7123 \pm .0114$ |
| CoreSet | $0.6933 \pm .0116$ | $0.6954 \pm .0194$ | $0.6996 \pm .0234$ | $0.7014 \pm .0206$ | $0.7036 \pm .0173$ | $0.7037 \pm .0200$ |
| ASGN | $0.6940 \pm .0070$ | $0.6999 \pm .0173$ | $0.6980 \pm .0206$ | $0.6984 \pm .0177$ | $0.6989 \pm .0169$ | $0.6999 \pm .0160$ |
| WAAL | $0.6587 \pm .0121$ | $0.6849 \pm .0202$ | $0.6936 \pm .0200$ | $0.6973 \pm .0200$ | $0.6960 \pm .0209$ | $0.6991 \pm .0171$ |
| EADA | $0.6311 \pm .0119$ | $0.6558 \pm .0170$ | $0.6713 \pm .0170$ | $0.6810 \pm .0175$ | $0.6865 \pm .0175$ | $0.6899 \pm .0145$ |
| Badge | $0.6958 \pm .0086$ | $0.7019 \pm .0119$ | $0.7054 \pm .0150$ | $0.7080 \pm .0095$ | $0.7096 \pm .0103$ | $0.7125 \pm .0097$ |
| MMPQ (Ours) | $0.6954 \pm .0073$ | $0.7198 \pm .0121$ | $0.7262 \pm .0125$ | $0.7302 \pm .0123$ | $0.7302 \pm .0103$ | $0.7293 \pm .0112$ |

|  | 6 | 7 | 8 | 9 | 10 |
|---|---|---|---|---|---|
| Random | $0.7065 \pm .0184$ | $0.7107 \pm .0198$ | $0.7123 \pm .0177$ | $0.7144 \pm .0211$ | $0.7165 \pm .0216$ |
| Entropy | $0.7122 \pm .0184$ | $0.7135 \pm .0152$ | $0.7146 \pm .0152$ | $0.7161 \pm .0154$ | $0.7190 \pm .0159$ |
| MC-Dropout | $0.7128 \pm .0132$ | $0.7142 \pm .0124$ | $0.7142 \pm .0126$ | $0.7161 \pm .0115$ | $0.7164 \pm .0109$ |
| CoreSet | $0.7054 \pm .0177$ | $0.7072 \pm .0170$ | $0.7090 \pm .0193$ | $0.7111 \pm .0188$ | $0.7110 \pm .0164$ |
| ASGN | $0.7060 \pm .0162$ | $0.7101 \pm .0155$ | $0.7155 \pm .0138$ | $0.7196 \pm .0119$ | $0.7225 \pm .0152$ |
| WAAL | $0.7001 \pm .0199$ | $0.7019 \pm .0159$ | $0.7021 \pm .0185$ | $0.7063 \pm .0166$ | $0.7055 \pm .0183$ |
| EADA | $0.6929 \pm .0146$ | $0.6943 \pm .0136$ | $0.6957 \pm .0122$ | $0.6971 \pm .0155$ | $0.6970 \pm .0157$ |
| Badge | $0.7143 \pm .0103$ | $0.7153 \pm .0091$ | $0.7153 \pm .0092$ | $0.7155 \pm .0107$ | $0.7159 \pm .0100$ |
| MMPQ (Ours) | $0.7294 \pm .0113$ | $0.7291 \pm .0109$ | $0.7294 \pm .0103$ | $0.7300 \pm .0099$ | $0.7302 \pm .0077$ |

**Table 6:** Numerical results of Fig. 4 (a). "Round 0" corresponds to performance on initial labeled pools.

|  | 0 | 1 | 2 | 3 | 4 | 5 |
|---|---|---|---|---|---|---|
| MMPQ | $0.6954 \pm .0073$ | $0.7198 \pm .0121$ | $0.7262 \pm .0125$ | $0.7302 \pm .0123$ | $0.7302 \pm .0103$ | $0.7293 \pm .0112$ |
| U.O. | $0.6943 \pm .0077$ | $0.7074 \pm .0116$ | $0.7126 \pm .0127$ | $0.7172 \pm .0111$ | $0.7208 \pm .0113$ | $0.7216 \pm .0102$ |
| D.O. | $0.6947 \pm .0069$ | $0.7218 \pm .0104$ | $0.7192 \pm .0110$ | $0.7197 \pm .0088$ | $0.7193 \pm .0085$ | $0.7192 \pm .0106$ |

|  | 6 | 7 | 8 | 9 | 10 |
|---|---|---|---|---|---|
| MMPQ | $0.7294 \pm .0113$ | $0.7291 \pm .0109$ | $0.7294 \pm .0103$ | $0.7300 \pm .0099$ | $0.7302 \pm .0077$ |
| U.O. | $0.7239 \pm .0075$ | $0.7269 \pm .0084$ | $0.7274 \pm .0090$ | $0.7282 \pm .0091$ | $0.7288 \pm .0098$ |
| D.O. | $0.7191 \pm .0127$ | $0.7184 \pm .0106$ | $0.7177 \pm .0097$ | $0.7169 \pm .0112$ | $0.7160 \pm .0105$ |

**Table 7:** Numerical results of Fig. 5. "Round 0" corresponds to performance on initial labeled pools.

|  | 0 | 1 | 2 | 3 | 4 | 5 |
|---|---|---|---|---|---|---|
| $(\bar{\mathbf{z}}_L, 0, 100)$ | $0.6922 \pm .0089$ | $0.7176 \pm .0115$ | $0.7252 \pm .0153$ | $0.7291 \pm .0103$ | $0.7292 \pm .0121$ | $0.7291 \pm .0138$ |
| $(\bar{\mathbf{z}}_F, 0, 100)$ | $0.6941 \pm .0097$ | $0.7189 \pm .0115$ | $0.7245 \pm .0151$ | $0.7293 \pm .0141$ | $0.7296 \pm .0132$ | $0.7299 \pm .0126$ |
| $(\bar{\mathbf{z}}_U, 0, 50)$ | $0.6942 \pm .0085$ | $0.7217 \pm .0108$ | $0.7282 \pm .0106$ | $0.7283 \pm .0106$ | $0.7285 \pm .0111$ | $0.7292 \pm .0110$ |
| $(\bar{\mathbf{z}}_U, 0, 500)$ | $0.6940 \pm .0096$ | $0.7193 \pm .0140$ | $0.7248 \pm .0162$ | $0.7287 \pm .0113$ | $0.7281 \pm .0125$ | $0.7294 \pm .0123$ |
| $(\bar{\mathbf{z}}_U, 1, 100)$ | $0.6923 \pm .0105$ | $0.7208 \pm .0119$ | $0.7292 \pm .0138$ | $0.7301 \pm .0126$ | $0.7298 \pm .0124$ | $0.7293 \pm .0124$ |

|  | 6 | 7 | 8 | 9 | 10 |
|---|---|---|---|---|---|
| $(\bar{\mathbf{z}}_L, 0, 100)$ | $0.7299 \pm .0150$ | $0.7295 \pm .0126$ | $0.7291 \pm .0121$ | $0.7298 \pm .0092$ | $0.7294 \pm .0133$ |
| $(\bar{\mathbf{z}}_F, 0, 100)$ | $0.7290 \pm .0125$ | $0.7299 \pm .0124$ | $0.7293 \pm .0106$ | $0.7294 \pm .0122$ | $0.7291 \pm .0122$ |
| $(\bar{\mathbf{z}}_U, 0, 50)$ | $0.7290 \pm .0091$ | $0.7294 \pm .0093$ | $0.7293 \pm .0110$ | $0.7294 \pm .0092$ | $0.7294 \pm .0091$ |
| $(\bar{\mathbf{z}}_U, 0, 500)$ | $0.7298 \pm .0124$ | $0.7294 \pm .0115$ | $0.7292 \pm .0137$ | $0.7298 \pm .0119$ | $0.7295 \pm .0102$ |
| $(\bar{\mathbf{z}}_U, 1, 100)$ | $0.7299 \pm .0119$ | $0.7287 \pm .0105$ | $0.7297 \pm .0111$ | $0.7300 \pm .0108$ | $0.7305 \pm .0114$ |

### A.8 Generalization to categorical classification or regression

Recall that, in MMPQ, we use $\max_{\hat{\mathbf{y}}} p(\hat{\mathbf{y}}|G)$ as a metric of model uncertainty. It is easy to see that, in the binary classification case, selecting those with small $\max_{\hat{\mathbf{y}}} p(\hat{\mathbf{y}}|G)$ is equivalent to selecting those with a large prediction entropy, which is a widely-used uncertainty metric. However, this may not hold in categorical classification or regression tasks.

We show in this subsection that, by choosing the KL-divergence between the prediction distribution and the uniform distribution as the uncertainty metric, we can generalize our approach to categorical classification or regression tasks. For simplicity, we take single-task categorical classification as an example.

We start with categorical classification. Suppose that the are $C$ classes in total, and denote $p_c = p(\hat{y} = c|G)$ ($c \in \mathcal{C} \triangleq \{1, \cdots, C\}$). The KL-divergence between the prediction distribution $p(\hat{y}|G)$ and a uniform distribution is

$$
\begin{aligned}
D_{\mathrm{KL}} &= \sum_{c=1}^{C} p_c \log(\frac{p_c}{\frac{1}{C}}) \\
&= \sum_{c=1}^{C} p_c (\log(p_c) + \log(C)) \\
&= \log(C) - (-\sum_{c=1}^{C} p_c \log(p_c)) \\
&= \log(C) - H(p),
\end{aligned}
\tag{20}
$$

where $H(p)$ is the entropy of the prediction distribution.

Since $\log(C)$ is a constant, selecting those with a small $D_{\mathrm{KL}}$ is equivalent to selecting those with a large $H(p)$. This shows that, $D_{\mathrm{KL}}$ is equivalent to $H(p)$ in terms of measuring model uncertainty.

Then, by replacing $\max_{\hat{\mathbf{y}}} p(\hat{\mathbf{y}}|G)$ with $H(p)$ in Eqn. 5, we obtain a new selection score:

$$
\begin{aligned}
H(p)p_L(G) &= -\left(\sum_{c=1}^{C} p_c \log(p_c)\right) p_L(G) \\
&= -\sum_{\hat{y}\in\mathcal{C}} q(G,\hat{y}) \log(p(\hat{y}|G)) \\
&= -\sum_{\hat{y}\in\mathcal{C}} q(G,\hat{y}) \log\left(\frac{q(G,\hat{y}')}{\sum_{\hat{y}'\in\mathcal{C}} q(G,\hat{y}')}\right) \\
&\propto -\sum_{\hat{y}\in\mathcal{C}} \exp(-E(G,\hat{y})) \log\left(\frac{\exp(-E(G,\hat{y}'))}{\sum_{\hat{y}'\in\mathcal{C}} \exp(-E(G,\hat{y}'))}\right).
\end{aligned}
\tag{21}
$$

The above equation shows that the selection score can also be computed using the EBM.

With the same reasoning, the selection score for regression tasks is:

$$
-\int_{\hat{y}\in Y} \exp(-E(G,\hat{y})) \log\left(\frac{\exp(-E(G,\hat{y}'))}{\int_{\hat{y}'\in Y} \exp(-E(G,\hat{y}'))\,\mathrm{d}\hat{y}'}\right) \mathrm{d}\hat{y},
\tag{22}
$$

where $Y$ is the support set.

Note that, the above formulation requires computing the integration over $Y$, which is intractable. This might be approximated by sampling and summation.

### A.9 Main results with unified y-axis scale

For better comparison, the Fig. 8 and Fig. 9 show the results of Fig. 2 and Fig. 3 with the same y-axis scale for each dataset.

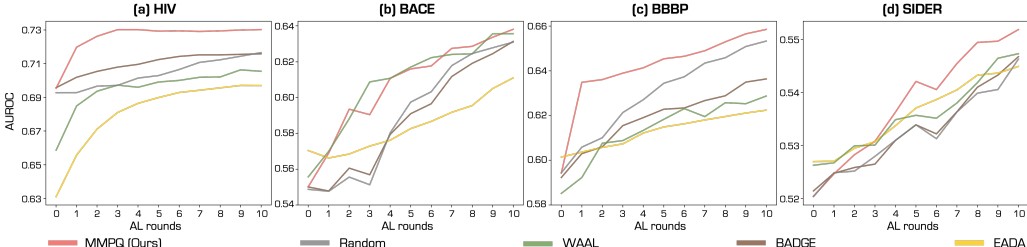

**Figure 8:** Active learning performance of MMPQ (ours) and baseline hybrid methods. "Round 0" corresponds to the performance on initial labeled pool.

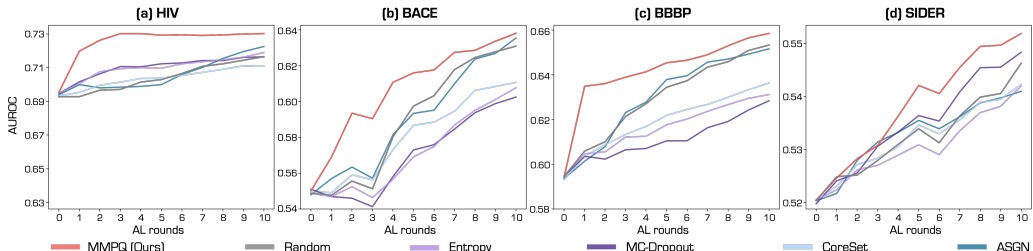

**Figure 9:** Active learning performance of MMPQ (ours) and uncertainty-based or diversity-based methods. "Round 0" corresponds to the performance on initial labeled pool.

