# OpenReview forum: "Jointly Modelling Uncertainty and Diversity for Active Molecular Property Prediction"
_logconference.io/LOG/2022/Conference — LoG 2022 Poster_

### Official Review · Reviewer_Yo8R · 2022-09-29

**Overall Score:** 5
**Confidence:** 4

**Review:**

Summary
=====
An energy-based model method for active learning is presented, combining estimates of diversity and uncertainty about model predictions in a single objective. Next candidates for queries are selected by a min-max strategy on that combined estimate score.
Experiments on a number of molecular property prediction datasets show performance matching or improving on a selection of baselines.

Strengths & Weaknesses
=====
* (+) Active learning for predicting molecular properties is indeed an important ingredient for more automated drug discovery pipelines.
* (~) This paper is not specific to graphs in any way.
* (-) The experiments show only limited advantages (improvements are very small) and have no uncertainty estimates (given that the experiment setup requires sampling, this is a substantial issue).

Overall, I think this paper is not quite ready for acceptance.

Questions
=====
* l187-l190: you're assuming conditional independence of labels - do you have evidence that this is a suitable assumption in the molecular property prediction space? In particular, I'm thinking of issues such as activity on-target and off-target, which is obviously not independent.
* Sect 4.2.1: have you experimented with a pre-trained autoencoder model (such as CDDD)? It's unclear to me if training a model on the (comparably small) set of labeled/unlabeled molecules is helpful or not, and so this would be an interesting experiment.
* How do you envision your system would be used in a realistic drug discovery project? The existence of a prepared set of likely unlabeled candidates is an artefact of the experiment setup on historical data - in reality, no such set exists. How would you construct a replacement for $\mathcal{D}_U$?

---

### Official Review · Reviewer_t74W · 2022-10-16

**Overall Score:** 5
**Confidence:** 3

**Review:**

**Summary**:
This paper proposes a new active learning method that optimizes joint distribution instead of posterior probability only, which naturally adds diversity to consideration. The idea is straightforward, and the method doesn’t rely on a manually defined trade-off between diversity and uncertainty. To make it work, the author combined it with an auto-encoder model and an energy-based model. The most significant flaw of this paper is the questionable experiments. Overall, I recommend accepting if the authors could add more thorough experiments to validate their method.

**Pro**:
- Correct and simple idea: The addition of $p(x)$ in the previous $p(y|x)$ to add diversity into consideration makes sense. The idea is simple, but if it works, we can expect more robust performance.

**Con**:
- Questionable empirical performance: The active learning methods always have a large variance in performance. However, from Figure 2, it seems like the authors are not using the same batch of the initially labeled pool to test all baselines. I recommend authors fix the initial batch of labeled data for all methods and add error bars for Figure 2-5 to validate their method.
- Is the EBM necessary?: Authors trained an auto-encoder to transform molecules into vectors in Euclidean space to apply an EBM model and estimate the probability of molecules. However, an auto-encoder model could estimate $p(x)$ by itself already. The authors should explain why they add another EBM.
- Limited technical contribution: The proposed method is simply an edition of existing methods, and the overall contribution is not gigantic.


**Other Comments**:
I recommend the authors explain the difference between their method and [1].

Reference:
[1] Ebrahimi, S., Gan, W., Chen, D., Biamby, G., Salahi, K., Laielli, M., ... & Darrell, T. (2020). Minimax active learning. arXiv preprint arXiv:2012.10467.

---

### Official Review · Reviewer_MmTW · 2022-10-21

**Overall Score:** 8
**Confidence:** 4

**Review:**

##########################################################################

Summary:

The authors propose a new active learning method for molecular properties that models a joint distribution and thus eliminates the need to model uncertainty and diversity separately with hyperparameters to balance them. They benchmark their method on four classification tasks from the MoleculeNet datasets and show that it has superior performance in batch-mode, pool-based active learning.

##########################################################################

Reasons for score (6):

The paper is a good contribution on the topic of active learning for molecules and addresses the issue of tradeoffs between uncertainty and diversity. The choices of experiments/benchmarks are logical and appropriate for the topic, but some form of cross validation is necessary to be able to draw more meaningful conclusions about how the new method compares to baselines.

##########################################################################

Strengths:

1. The paper does a good job of discussing prior active learning strategies and explaining the distinguishing features of the new strategy proposed here.

2. The novel objective function and the method of maximizing q by varying y for a single G, then selecting the batch of G that minimizes the max(q) is intuitive.

3. The use of scaffold splitting rather than random splitting allows for better testing of reproducibility.

4. The ablation study and subsequent analysis is very informative, and the conclusion that diversity is more helpful in earlier iterations while the model "doesn't know what it doesn't know" is logical and aligns with prior literature on the topic.

##########################################################################

Weaknesses / major questions:

1. (Line 75) The word "significantly" is used to describe the gain in performance over baseline active learning methods. Since there was no cross-validation (e.g. using a few random seeds to randomly sample several different starting sets of 10% of the training data) to get error bars on the results, I don't think it's appropriate to use this word here (this should be reserved for *statistical* significance). While the performance gains seem large for some of the datasets, stronger conclusions could be drawn if there were error bars on the curves in Figures 2 and 3.

2. How were the hyperparameters selected for the baseline methods that required hyperparameters to balance the uncertainty and diversity? Were the defaults used, or the best for each task?

##########################################################################

Minor questions / suggestions:

1. (Line 46) For models that require tuning a hyperparameter to tradeoff between uncertainty and diversity, this tuning can often be done in a retrospective analysis (i.e. on an existing dataset with 0 annotation cost) before use in a prospective setting where annotation cost > 0. I think this work has an advantage over the retrospective tuning since the retrospective approach may not be representative of the prospective setting, but maybe this caveat would be worth mentioning.

2. (Lines 69, 201) It sounds a bit odd to me to refer to a SMILES string as an "expert-defined sequence" (as if they're similar to molecular descriptors chosen by chemists), and I don't usually see them described this way.


3. (Line 188 / Apps. A.2) Is conditional independence a good assumption for the case of the SIDER dataset? It's certainly a useful assumption since it makes the computation tractable, but it seems like it would be far from true for the 27 organ classes in SIDER. Could this be part of the reason why the new AL strategy does not seem to have as large of performance gains for SIDER in Figures 2 and 3?

4. (Line 247) "Among them, ASGN is the only existing method that investigates AL in molecular property prediction." I think this is true based on the implementations you cite, but may not be true about these methods in general (e.g. MC-dropout is included in active learning benchmarks for molecules here: https://doi.org/10.1021/acscentsci.1c00546).

5. (Figures 2 and 3) I think it would be helpful to have consistent y-axis limits for each task between Figures 2 and 3 (i.e. HIV and BACE don't need to have the same limits, but maybe Figure 2 HIV and Figure 3 HIV should) since the MMPQ curve is the same in each figure. This is almost the case already for parts b, c, and d, but part a is quite different between Figures 2 and 3.

6. I would love to see something like this benchmarked on regression datasets with other common uncertainty quantification methods (e.g. evidential, ensemble, mean-variance estimation, etc.), but that is outside of the scope of this paper and shouldn't be required for acceptance to this conference.

#########################################################################

Typos:

1. (Line 341) "train" -> "training"


#########################################################################

---

### Official Review · Reviewer_WMPe · 2022-10-26

**Overall Score:** 8
**Confidence:** 3

**Review:**

**Summary**:
The paper proposes an acquisition strategy for acquiring unlabeled data from a given pool of unlabeled samples in an Active Learning (AL) setting on a set of binary classification tasks. The proposed strategy -- Minimum Maximum Probability Querying (MMQP) -- is a hybrid strategy, i.e. it takes into account *uncertainty* and *diversity* to choose data points to label, but in contrast to most other hybrid models eliminates the need to tune hyperparameters. It is based on energy based models (EBMs), which model the joint distribution. The paper evaluates the proposed MMQP strategy on several benchmark datasets for molecular property prediction and find that it outperforms other *uncertainty*, *diversity* or *hybrid* approaches consistently.

*Contributions*:

- *[C1]* The paper proposes a new acquisition strategy (MMQP) that balances uncertainty and diversity objectives without requiring hyperparameter tuning. The discussion in the paper theoretically motivates the strategy and provides intuition via an ablation study.
- *[C2]* In contrast to most papers looking into AL acquisition strategies, which typically work with image data, this paper looks at graph based input data (SMILES strings of molecules) and deals with the problem of graph data working in an embedding space.
- *[C3]* The proposed MMQP strategy is evaluated against comparable uncertainty based, diversity based and hybrid AL schemes on multiple molecular property prediction datasets and includes an ablation study of the proposed MMQP method.

-------------------------------------------------------

**Strengths & Weaknesses**:

*Strengths*:

- *[S1]* Well presented: The paper is clearly written and well-structured. Where questions pop-up, authors pre-emptively address them. To give two examples: E.g. at figure 2 they explain why the baseline methods *EADA* and *WAAL* start at lower AUROC's, and in line 187 onwards they comment on why the seeming inefficiency of their model, which at first sight seems to require summation over all $2^n$ options, can be circumvented. Explanations such as these show that the paper was written with the reader in mind and thought was put into the presentation. I enjoyed reading it!
- *[S2]* Consistently outperforms or matches benchmarks over 4 tasks: The proposed method consistently outperforms all benchmark strategies (*uncertainty*, *diversity* and *hybrid* ones) or matches the performance of the best benchmark.
- *[S3]* Provides intuition for the modelling approach and performs meaningful ablations: The paper presents theoretical reasoning (see section 4.1.1) to motivate the proposed acquisition strategy and give an intuition as to why design choices were made. It also performs an ablation study to further improve this intuition.

*Weaknesses*:

- *[W1]* The reference to *molecular property prediction* in the title lead me to assume that we might also look at non-binary classification or regression tasks. However, it became apparent only later in the paper that we focus on a multitask binary prediction setting. It would be helpful for the reader if this was made clear earlier on and in the abstract.
- *[W2]* There is currently little discussion of the limitations of the proposed approach. For example, using $\text{max}$ as summary statistic to describe the *uncertainty* of the distribution $p(\hat{y}|G)$ works well for the binary classification setting (Bernoulli distributions) tackled in this paper. But the explanation given in section 4.1.1 (c.f. line 165-166) does not carry over to a multiclass (categorical) setting. It would be great if the authors could point out and discuss limitations such as this one in a dedicated section and potentially give an outlook on tackling them. As an alternative summary statistic, one could e.g. use the Kullback-Leibler divergence of the predicted label probability distribution to a uniform distribution.
- *[W3]* The evaluation currently does not perform any error analysis or cross validation.

-------------------------------------------------------

**Recommendation**:
Clear accept. It was a pleasure to read the paper, and it gave me lots of new ideas. I think it will be of value to the research community.

-------------------------------------------------------

**Questions for clarification, improving the paper quality and out of curiosity**:

- *[Q1]* How would you generalize the proposed model to the case of categorical labels or to a regression setting. Instead of $\text{max}$, could you use KL divergence to the uniform distribution to characterize *uncertainty*?
- *[Q2]* What are some limitations of the discussed approach? For a large pool of unlabeled data, potentially many evaluations of the model are needed - any ideas on how to best tackle this? (e.g. as a realistic setting in drug discovery one would often want to find new compounds rather than just work with a set of pre-defined graphs in a given pool)
- *[Q3]* The introduction mentions that some baselines are highly variable with respect to hyperparameter settings. Could you elaborate on how you chose those hyperparameters for the benchmark evaluation?
- *[Q4]* It seems that the `random` method performs better than several baselines on many of the tasks. Do you have an explanation for why that is the case?
- *[Q5]* Could you add a note to figures 2 and 3 to indicate that the y-axis is offset?
- *[Q6]* Could you add uncertainty estimates/CV to your evaluation?

-------------------------------------------------------

**Type of paper**: Full paper proceedings track submission.

---

### Meta-Review · Area_Chair_uk3a · 2022-11-15

**Confidence:** 3
**Recommendation:** Accept

**Meta Review:**

This paper developed an active learning framework for molecular property prediction (binary classification setting). Currently, two reviewers vote for weak rejection while the other two vote for strong acceptance. The main concern from the reviewers is the lack of error bars/cross validation, small improvement over baselines, lack of technical contribution, and questions on the necessity of energy based models. During the discussion stage, authors have revised their manuscript to report standard deviation of all methods under cross validation. The improvements are relatively small on some datasets (SIDER) but reasonable on other datasets (HIV). Energy based model is not clear but seems a reasonable choice for density estimation. I believe that the MMPQ strategy is different from previous work (Minimax active learning) and modeling the diversity term via density estimation is a reasonable contribution to the community. Despite some disagreement between the reviewers, the area chair believes that authors have addressed concerns and the paper can be accepted.

---

### Decision · Program_Chairs · 2022-11-22

Accept (Poster)